# Patients with nodding syndrome in Uganda improve with symptomatic treatment: a cross-sectional study

Richard Idro,[1,2] Hanifa Namusoke,[1] Catherine Abbo,[3,4] Byamah B Mutamba,[3,5] Angelina Kakooza-Mwesige,[1,6] Robert O Opoka,[1] Abdu K Musubire,[7] Amos D Mwaka,[7] Bernard T Opar[8]

For numbered affiliations see end of article.

**Correspondence to**
Dr Richard Idro; ridro1@
gmail.com

## ABSTRACT

**Objectives:** Nodding syndrome (NS) is a poorly understood neurological disorder affecting thousands of children in Africa. In March 2012, we introduced a treatment intervention that aimed to provide symptomatic relief. This intervention included sodium valproate for seizures, management of behaviour and emotional difficulties, nutritional therapy and physical rehabilitation. We assessed the clinical and functional outcomes of this intervention after 12 months of implementation.

**Design:** This was a cross-sectional study of a cohort of patients with NS receiving the specified intervention. We abstracted preintervention features from records and compared these with the current clinical status. We performed similar assessments on a cohort of patients with other convulsive epilepsies (OCE) and compared the outcomes of the two groups.

**Participants:** Participants were patients with WHO-defined NS and patients with OCE attending the same centres.

**Outcome measures:** The primary outcome was the proportion of patients with seizure freedom ($\geq 1$ month without seizures). Secondary outcome measures included a reduction in seizure frequency, resolution of behaviour and emotional difficulties, and independence in basic self-care.

**Results:** Patients with NS had had a longer duration of symptoms (median 5 (IQR 3, 6) years) compared with those with OCE (4 (IQR 2, 6) years), p<0.001. The intervention resulted in marked improvements in both groups; compared to the preintervention state, 121/484 (25%) patients with NS achieved seizure freedom and there was a >70% reduction in seizure frequency; behaviour and emotional difficulties resolved in 194/327 (59%) patients; 193/484 (40%) patients had enrolled in school including 17.7% who had earlier withdrawn due to severe seizures, and over 80% had achieved independence in basic self-care. These improvements were, however, less than that in patients with OCE of whom 243/476 (51.1%) patients were seizure free and in whom the seizure frequency had reduced by 86%.

**Conclusions:** Ugandan children with NS show substantial clinical and functional improvements with symptomatic treatments suggesting that NS is probably a reversible encephalopathy.

## Strengths and limitations of this study

- This is the largest cohort of patients with nodding syndrome ever reported on to date. The report examines preintervention clinical and functional features before a well-designed treatment intervention was implemented and how these improved over the course of the ensuing year. The improvements in patients with nodding syndrome were also compared with those of a similar cohort with other epilepsies.
- However, we did not conduct a prospective study but rather before and after cross-sectional studies, meaning that we cannot comment on the incidence of death or loss to follow. We also relied on patient records for the preintervention features.
- Other than head nodding, seizures in nodding syndrome are similar to seizures in other convulsive epilepsies and over time, head nodding may cease in some patients with nodding syndrome, increasing the risk of misclassification.
- In addition, we did not determine compliance with antiepileptic drugs or have reports of the adverse effects patients experienced while on treatment and did not have a detailed documentation of the nutritional and cognitive stimulatory treatments each child received. We, however, limited the effects of such bias by choosing only a few outcome measures that are not easily confused.

## BACKGROUND

Nodding syndrome is a poorly understood devastating neurological disorder affecting several thousand children in the sub-Saharan African countries of South Sudan,[1–3] Uganda[4–6] and Tanzania.[7–9] The syndrome is characterised by the almost daily atonic seizures manifesting as clusters of head nods[4] and complicated by tonic clonic, focal, myoclonic and/or atypical absence seizures, cognitive and motor decline, malnutrition, behavioural and emotional difficulties.[6] [7] The aetiology is unknown, although the

syndrome has been associated with infestation with *Onchocerca volvulus*.[1] [5] [7] Studies of Tanzanian and Ugandan patients have concluded that nodding syndrome is probably symptomatic generalised epilepsy.[4] [6] [7]

In Uganda, a multidisciplinary team developed management guidelines for care.[10] The objective was to relieve symptoms, as well as to offer primary and secondary prevention of disability, and rehabilitation to improve function. The most important clinical needs were identified as seizure control, relief of behavioural and emotional difficulties, nutritional therapy, physical and cognitive rehabilitation. The first group of patients were enrolled in March 2012. We evaluated clinical outcomes of this intervention after a minimum of 12 months. We hypothesised that if treated with appropriate anticonvulsants, patients with nodding syndrome would achieve similar seizure control like patients with other convulsive epilepsies. We therefore, in addition, compared outcomes of patients with nodding syndrome with those in patients with other convulsive epilepsies.

## METHODS
### Design and setting
This was a cross-sectional survey of a cohort of patients with nodding syndrome that evaluated the clinical and functional outcomes of patients receiving the Ugandan Ministry of Health treatment intervention at least 12 months after initiation of therapy. We performed a similar evaluation on a cohort of patients with other convulsive epilepsies that attended the same centres and compared improvements in the two groups. The study was conducted in northern Uganda, the region most affected by nodding syndrome in the country. The population prevalence of probable nodding syndrome among children of the affected age group in the study area has been estimated as 6.8 (95% CI 5.9 to 7.7) per 1000.[11] This region also suffered a protracted armed rebellion that lasted over 20 years[12] resulting in massive internal displacement. It is only in the past 6–7 years that peace prevailed and the population returned to their homes.

### Participants
Participants were patients with either nodding syndrome or other convulsive epilepsies receiving treatment at any one of the nodding syndrome treatment centres in the seven districts of Lamwo, Kitgum, Pader, Gulu, Amuru, Lira and Oyam. The definition of head nodding and diagnosis of nodding syndrome is in accord with the criteria developed by international consensus during the WHO facilitated meeting on nodding syndrome in Kampala, 2012.[13] Head nodding was defined as repetitive, involuntary drops of the head on to the chest in previously normal persons. We included probable and confirmed cases only. Children with other convulsive epilepsies were those with active (at least one in the past year) tonic–clonic or focal jerking epileptic seizures. The diagnosis and classification of epilepsy in this rural

community is quite limited, and in many cases categorisation into specific clinical groups is not possible. We therefore only included those with convulsive epilepsies. Participants with onset of symptoms outside of the ages 3–18 years were excluded to allow comparability with patients with nodding syndrome.

### The intervention
The nodding syndrome treatment centres in Lamwo, Kitgum and Pader were opened in March 2012 followed by those in Amuru, Gulu, Lira and Oyam in June 2012. Prior to this, clinicians and nurses at each centre underwent a 5-day training on the management of nodding syndrome using the specified guideline.[14] The training, which also included general principles of epilepsy treatment, was provided through didactic lectures, role play, bedside clinical teachings and demonstrations by the same team that developed the guidelines. At the end of the 5 days, each team returned to their centre and worked with the trainers to initiate provision of care. Other than the centre in Kitgum, which is a district hospital (a level V health centre), all the others were health centre III. At each centre, clinical service was led by a medical or psychiatric clinical officer (individuals with a diploma in clinical medicine or psychiatry after 3 years of training), general and psychiatric nurses, laboratory technicians and either a physiotherapist or occupational therapist. In Kitgum hospital, the team was led by a medical officer (MBChB). These teams were supported by local lay volunteers—village health workers—who coordinated follow-up and ambulatory care in homes. In each district, supervisory oversight was provided by a district nodding syndrome focal person, the District Health Officer and the district nodding syndrome committee, while nationally there was a national nodding syndrome coordinator who brought everyone together. Over the next 12 months, each centre received support supervision visits on at least two occasions to maintain skills and attend to issues arising.

Details of the treatment are described elsewhere.[10] In summary, inpatient emergency care was offered to patients with life-threatening comorbidities. Ambulatory and community care was offered to patients without comorbidities or those with non-life-threatening comorbidities. Sodium valproate was the first-line anticonvulsant starting at 10 mg/kg/day in two divided doses, and the dose titrated to a maximum of 40 mg/kg/day. The patient's family was provided with supplemental food rations every 2–4 weeks. Severely malnourished patients with medical complications were treated as inpatients and those with uncomplicated severe malnutrition were treated as outpatients with ready to use therapeutic feeds. This was provided as Plumpy'Nut, a product of Nutriset (Normandy, France). Plumpy'Nut is made of peanut paste, vegetable oil, powdered milk and sugar, vitamins (A, B-complex, C, D, E and K) and minerals (calcium, phosphorus, potassium, magnesium, zinc, copper, iron, iodine, sodium and selenium) all combined in a foil pouch. Each 92 g pack provides 500

calories. Management of behaviour and emotional difficulties included counselling and referral of those with severe symptoms to mental health services. Other management included physical, speech and language therapy and cognitive stimulation.

Children with other convulsive epilepsies were provided with first-line anticonvulsants (carbamazepine, phenobarbitone, phenytoin or sodium valproate) or continued to receive earlier prescribed anticonvulsants, but the dose was adjusted appropriately. A new anticonvulsant was introduced if an inappropriate drug was being provided. Anticonvulsants such as oxcarbazepine, lamotrigine, levetiracetam and topiramate are unavailable in the public health service in Uganda. Families of patients with other convulsive epilepsies were also provided with similar supplemental feeding. In addition, parents/carers of both groups of patients were educated on seizures, epilepsy, adherence to antiepileptic drugs and prevention of seizure-related injuries.

### Sample size
In a preliminary evaluation of the treatment outcomes of nodding syndrome after 7 months of intervention, we documented (from the parental or carer report) that 5/47 (10%) patients had achieved seizure freedom (no head nodding or convulsive seizures) for at least 30 days prior to the visit. Using these findings, we estimated that a sample of 432 patients will be sufficient at 5% level of significance and 90% power to detect a 10% increase in this proportion to 20% after 12 months of treatment. Second, up to 70% of children with new onset convulsive epilepsies achieve terminal seizure remission with drug treatment.[15] [16] The onset of seizure remission is often evident within the first year of treatment.[15] Using these findings, we estimated that with a sample of 461 patients with nodding syndrome and a similar number with other convulsive epilepsies, at 90% power and a 5% level of significance, we would be able to reject the null hypothesis that there is no difference in the proportions of patients with nodding syndrome or other convulsive epilepsies achieving seizure freedom with 12 month therapy. We set to recruit the larger sample.

### Study procedures and measurements
As of 30 June 2013, there were 3295 patients with probable or confirmed nodding syndrome receiving care at the seven centres. We used proportionate sampling to estimate the number of participants to be recruited from each centre and consecutively recruited patients as they presented until the sample was achieved. Data were collected between 1 July 2013 and 30 September 2013. One of two investigators (RI or BTO) first conducted a day's training on the study procedures followed by a joint clinic with the clinicians at the centre. The local clinical team subsequently worked independently until study completion. Case record forms were completed from data abstracted from preintervention records, direct inquiry from parents/carers and on physical examination. The preintervention seizure burden, weight and height, and behaviour or emotional difficulty were obtained from records. We defined seizures as head nodding or convulsive seizures and defined seizure burden as the number of clusters of head nodding and/or convulsive seizures per unit time.

In the clinic, parents/carers reported on current seizures, behaviour and emotional difficulties. Weight was measured using a stand-on electronic scale while height or length was measured using a stadiometer. Independence in basic self-care (self-feeding, dressing and using a toilet), the status of schooling and the ability to appropriately help with culturally and age-appropriate home care activities (eg, sweeping the compound) were obtained from the parents or carers. The parents and carers were also asked to provide an overall assessment of improvements or worsening of symptoms over the year on an ordinal scale (markedly improved, some improvement, no improvement or worse).

### Outcome measures
The primary outcome was the proportion of patients who had achieved seizure freedom (defined as ≥1 month without seizures (no head nodding and/or convulsive seizures observed by the parent/carer prior to the follow-up visit)). Secondary outcomes included a reduction in the seizure burden (reduction in the mean number of clusters of head nods and/or convulsive seizures per unit time), the proportions of patients with independence in basic self-care, resolution of behaviour and emotional difficulties and enrolment in school.

### Data management and statistical analysis
Data were collected on case record forms and double-entered into a Microsoft Access 2007 database. Data analysis was performed using STATA V.12.0 (STATA Corp, Texas, USA). The two patient groups were considered as two independent single samples and paired data (before initiation of therapy and at least 12 months later) analysis was performed for each group. Thus, we determined the proportions of patients with nodding syndrome with seizure freedom before and after 12 months and the proportions with the different secondary outcomes. A one sample t test was used to compare means of normally distributed continuous data, the Mann-Whitney U test for medians of skewed data and McNemar's test for categorical data. We then examined for patient characteristics potentially associated with seizure freedom including duration and age at onset of symptoms, baseline seizure frequency, presence of behaviour and emotional difficulties, whether the child had head nodding only or head nodding plus (other seizures) and antiepileptic drug dose and performed a logistic regression analysis to determine variables independently associated with achieving seizure freedom.

## RESULTS
### General descriptions

A total of 1322 participants were screened in six of the seven districts. Oyam district, which had only eight patients with nodding syndrome, was not visited. Two hundred and fifteen participants were ineligible. Another 147 were also excluded for different reasons. Thus, 960 participants (484 with nodding syndrome and 476 with other convulsive epilepsies) were available for the study (figure 1).

The two groups were of similar age and gender; the mean (SD) age of patients with nodding syndrome was 13.7 (3.6) years and that of patients with other convulsive epilepsies was 13.0 (2.9) years, p=0.998; 281/484 (58.1%) participants with nodding syndrome and 267/476 (56.1%) with other convulsive epilepsies were male, p=0.538. However, participants with nodding syndrome had experienced a longer duration of symptoms (median 5 (IQR 3, 6) years) compared to patients with other epilepsies, (median 4 (IQR 2, 6) years), p<0.001.

The median daily dose of sodium valproate in patients with nodding syndrome was 16 (IQR 12, 21) mg/kg/day with most (298/484, 61.6%) on relatively low doses (<20 mg/kg/day). The majority of the patients with other convulsive epilepsies (421/476, 88.5%) were on carbamazepine, phenobarbitone or phenytoin monotherapy. The remaining 55 were either on sodium valproate (40/476, 8.4%) or combinations of the above anticonvulsants (15/476, 3.1%).

### Outcomes of interventions
#### Seizures

There was a marked reduction in seizures with the intervention; overall, 25% (95% CI 21 to 29) of patients with nodding syndrome achieved seizure freedom. Both the frequency of head nodding and of convulsive seizures reduced by over 70%. The reduction in seizure burden was even more marked in patients with other convulsive epilepsies; 51% (95% CI 46.4 to 55.6) achieved seizure freedom and the overall burden of seizures decreased by 86%, (table 1).

Although the effects of sodium valproate on seizure control in nodding syndrome was evident at relatively low doses, additional patients achieved seizure freedom with dose escalation. Thus, 87/298 (29.2%) patients were seizure free on sodium valproate <20 mg/kg/day and an additional 34/186 (18.3%) patients achieved seizure freedom with dose increases to 20–40 mg/kg/day.

We repeated diagnostic EEG recordings for three patients with nodding syndrome who were part of the 22 we reported on earlier.[6] The recordings showed clear improvements in background EEG and reductions in previously widespread interictal epileptiform discharges. All three were on sodium valproate 20–25 mg/kg/day and were experiencing only occasional convulsive seizures but no head nodding.

### Behaviour and emotional difficulties

Behaviour and emotional difficulties were reported in 327 (67.6%) participants with nodding syndrome and in 250 (52.5%) with other convulsive epilepsies prior to the intervention. Among participants with nodding syndrome, these included aggressive and destructive behaviour (186/484, 39.5%), wandering or running away (113/484, 23.4%) and periods of low mood (114/484, 23.6%). Over the 12 months, the difficulties resolved in 194/327 (59.3%) patients with nodding syndrome and in 145/250 (58%) patients with other convulsive epilepsies. Improvements were most evident in patients with nodding syndrome initially reporting wandering, aggressive and destructive behaviour. Psychotropic drugs (haloperidol) were prescribed for only three patients with severe difficulties and two received anxiolytic drugs. An additional 62 (12.8%) patients with nodding syndrome, especially those with uncontrolled or worsening seizures, developed new onset behaviour and emotional difficulties; these included 44 (9.1%) with aggressive and destructive behaviour, 18 (3.7%) with wandering behaviour and 21 (4.3%) with mood problems. Wandering behaviour was uncommon among patients with other convulsive epilepsies in whom impulsive behaviour and hyperactivity were more common.

### Independence in basic self-care

Prior to the intervention, 174/484 (36%) patients with nodding syndrome were independent in basic self-care. This proportion had increased to 402/484 (83.1%) patients by the time of the survey, p<0.001. Similar improvements were observed in patients with other convulsive epilepsies. Thus, 397/476 (83.4%) patients were independent in basic self-care at the time of the survey, up from 270/476 (56.7%) patients prior to the intervention, p<0.001.

### School attendance

A total of 443 patients (193/484, 39.9% with nodding syndrome and 250/476, 52.5% with other convulsive epilepsies) were enrolled in and attending school at the time of the survey. This included 86/484 (17.8%) patients with nodding syndrome and 80/476 (16.8%) patients with other convulsive epilepsies who had returned to school with seizure control and improvements in other symptoms. Although these children had returned to school, parents reported that 90/193 (46.6%) patients with nodding syndrome and 76/250 (30.4%) patients with other epilepsies were still performing poorly in school.

### Qualitative assessment of improvements by parents and carers

On an ordinal subjective scale, parents felt that 112/484 (23.1%) patients with nodding syndrome and 253/476 (53.2%) patients with other convulsive epilepsies had improved markedly. Another 325/484 (67.2%) patients with nodding syndrome and 194/476 (40.8%) patients with other convulsive epilepsies had some improvement. The number of patients with nodding syndrome who could participate and help their parents with home care

**Figure 1** Participant recruitment.

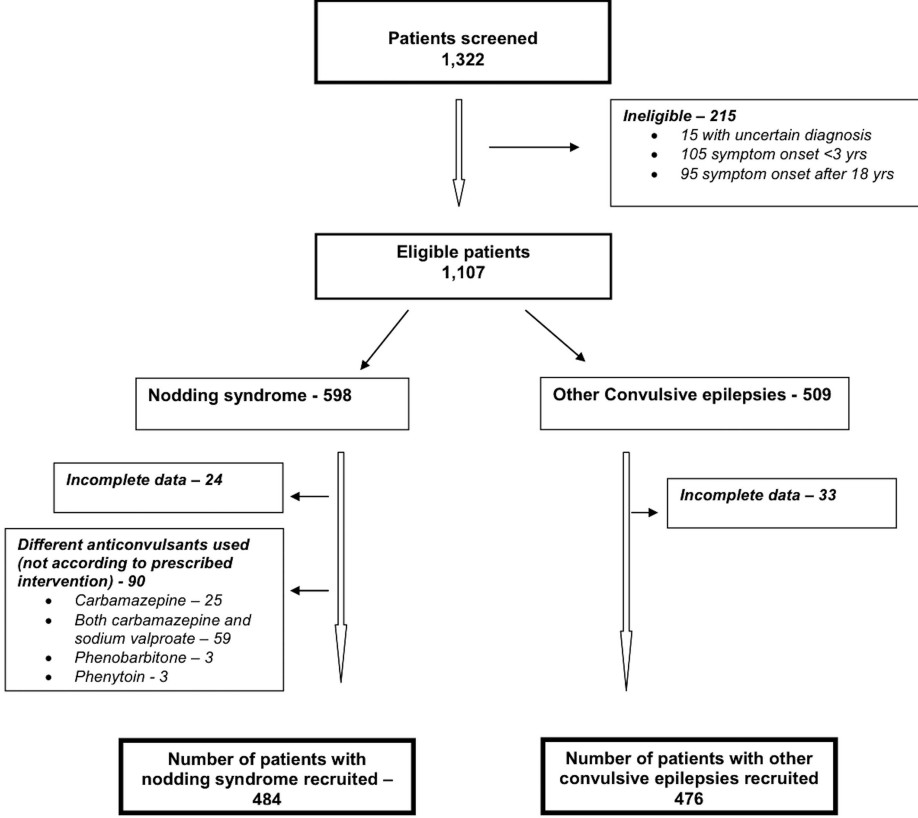

tasks increased from 152/484 (31.4%) to 372/484 (76.9%) with the intervention. Only 47/484 (9.7%) patients with nodding syndrome and 29/476 (6.1%) patients with other convulsive epilepsies had no improvement in symptoms or became worse over the period of intervention.

### Prognostic factors for seizure freedom

While we examined the relationship between gender, age at onset of symptoms, duration of symptoms, baseline seizure frequency, presence of behaviour and emotional difficulties, whether the child had head nodding only or head nodding plus (other seizures), antiepileptic drug dose and achieving seizure freedom, only a lower number of clusters of head nods prior to the intervention (adjusted OR 0.80 (95% CI 0.72 to 0.88), p<0.001) and response to a lower antiepileptic drug dose (adjusted OR 0.96 (95% CI 0.93 to 0.99), p=0.046) were independently associated with achieving seizure freedom.

### DISCUSSION

Our study aimed to determine the clinical outcomes and the effectiveness of a symptomatic treatment intervention for nodding syndrome. We documented substantial clinical and functional improvements with the intervention. The findings suggest that nodding syndrome is probably a reversible encephalopathy. The improvements we observed were, however, less than that

seen in patients with other convulsive epilepsies, suggesting that epileptic seizures in nodding syndrome may be less anticonvulsant sensitive compared to seizures in the other convulsive epilepsies.

Although the proportion of patients with nodding syndrome who achieved seizure freedom was modest, our findings suggest that a treatment package of selected anticonvulsants, psychobehavioural interventions and nutritional and physical rehabilitation can control seizures, improve function and even reverse some severe functional disability in nodding syndrome. This observation seems to concur with a report from Tanzania in which although seizure freedom was achieved by 2/32 patients treated with phenobarbitone, over 80% had reductions in seizure burden.[9] Even though we did not perform specific cognitive testing or brain imaging to objectively document functional and structural improvements with the intervention, comparisons of a preintervention and repeat EEG recordings in three patients with previous recordings demonstrated clear improvements in background EEG and reductions in the previously widespread interictal epileptiform discharges.[6]

Clinical trials comparing treatment of seizures in nodding syndrome with sodium valproate to treatment with other anticonvulsants such as lamotrigine or levetiracetam either as monotherapy or as an add-on therapy may be considered. In addition and especially for patients whose symptoms are either not controlled or became worse on therapy, other strategies may be considered. Epileptic encephalopathy is a possibility,

**Table 1** Preintervention features and features at least 12 months after initiation of a symptomatic treatment intervention in patients with nodding syndrome or other convulsive epilepsies

| | Patients with nodding syndrome, N=484 | | | Other convulsive epilepsies, N=476 | | |
|---|---|---|---|---|---|---|
| | Preintervention status | Features ≥12 months later | p Value | Preintervention status | Features ≥12 months later | p Value |
| Patients with seizure freedom*, % Seizure free for longer than one month | 8 (2%) (95% CI 0.07 to 3.2) | 121 (25.0%) (95% CI 21.2 to 29.1) | <0.001 | 8 (2%) (95% CI 0.7 to 3.3) | 243 (51.1%) (95% CI 46.4 to 55.6) | <0.001 |
| Daily clusters of head nods, median (IQR) | 4 (IQR 3, 6) | 1 (IQR 0, 2) | <0.001 | – | – | – |
| Patients with behaviour and emotional difficulties, % | 327/484 (67.6%) (95% CI 63.2 to 71.7) | 133 (27.5%) (95% CI 23.5 to 33.7) | <0.001 | 250/476 (52.5%) (95% CI 47.9 to 57.1) | 105 (22.1%) (95% CI 18.4 to 26.1) | <0.001 |
| GMFCS score† | | | | | | |
| 1 | 185/282 (64.0%) | 223/282 (79.1%) | <0.001‡ | 212/288 (73.6%) | 239/288 (83.0%) | <0.001‡ |
| 2 | 58/282 (20.1%) | 39/282 (13.8%) | | 41/288 (14.1%) | 39/288 (13.5%) | |
| 3 | 39/288 (13.5%) | 39/288 (13.5%) | | 39/288 (13.5%) | 10/288 (3.5%) | |
| 4 and 5 | 39/288 (13.5%) | 0 (0) | | 2/288 (0.7%) | 0 (0) | |
| Independence in basic self-care, % | 174 (36.0%) (95% CI 31.7 to 40.4) | 402 (83.1%) (95% CI 79.4 to 86.3) | <0.001 | 206 (43.3%) (95% CI 38.8 to 47.9) | 397 (83.4%) (95% CI 79.8 to 86.6) | <0.001 |
| Able and performs culturally and age appropriate home care activities, % | 152 (31.4%) (95% CI 27.2 to 37.4) | 372 (76.9%) (95% CI 72.8 to 80.5) | <0.001 | 187 (39.3%) (95% CI 34.9 to 43.8) | 382 (80.3%) (95% CI 76.4 to 83.7) | <0.001 |
| Enrolled at and attending school, % | 107 (22.1%) (95% CI 18.5 to 26.1) | 193 (39.9%) (95% CI 35.5 to 44.4) | <0.001 | 170 (35.7%) (95% CI 31.4 to 40.2) | 250 (52.5%) (95% CI 47.9 to 57.1) | <0.001 |

*≥1 Month without seizures.
†GMFCS=Gross Motor Function Classification Score; N=282; that is, only 282 patients with nodding syndrome had paired GMFCS pre and post interventions scores obtained.
‡$\chi^2$ Test for trend with Yate's correction.

especially in patients with severe and persistent symptoms. Can therapy with benzodiazepines, high-dose steroids or other immunosuppressant drugs be considered?[17]

The aetiology of nodding syndrome is still unknown. In all three countries where nodding syndrome has been described, it has been associated with infestation by *Onchocerca volvulus*.[1 7 18] Uganda is in its second year of twice yearly mass administration of ivermectin (an antimicrofilarial agent active only against the mirofilaria but not the adult parasite). Other strategies that target both microfilaria and the adult worms and/or their cosymbiotic bacteria, *Wolbachia*, may be considered as potential specific therapy.[19 20]

Despite these improvements, parents reported that the majority of the 40% of children who returned to school continued to perform poorly. There is a need to examine whether the continued poor academic performance is due to irreparable brain injury or an underlying ongoing aetiopathogenic process. To date, there are no systematic studies of cognitive function in nodding syndrome. Such studies will help define areas of functional deficits and document improvements preferably using tools that can be applied across different regions with minimal modification to allow comparison.

We did not apply specific psychiatric diagnostic tools to patients with behaviour and emotional difficulties to be able to make distinct psychiatric diagnoses. A few children with severe difficulties were attended to by the local mental health services and some were given psychotropic drugs. The majority of the 194 children in whom behaviour and emotional difficulties resolved, however, improved without psychotropic drugs but with seizure control, suggesting that in nodding syndrome some of these features may be comorbidities of epilepsy. Wandering behaviour may be an ictal event.[6] Some patients may also have benefited from the effects of sodium valproate on behaviour; in a recent case series of Ugandan children, Musisi *et al*[21] documented improvements in some patients receiving antidepressants. Put together, these findings suggest that psychotropic drugs may be considered for some patients with nodding syndrome, especially those with severe symptoms.

### Study limitations

First, other than head nodding, seizures in nodding syndrome may be similar to seizures in other convulsive epilepsies and over time, head nodding may cease in some patients.[9] This scenario opens room for potential misclassification of disease as the current disease criteria heavily leans on clinical observations. Second, we did not perform a prospective study; instead, we relied on patient records for preintervention features. Third, we had only limited data on the burden and severity of other comorbidities such as injuries (eg, burns) or earlier exposure to acute encephalopathies such as cerebral malaria, meningitis and encephalitis. Fourth,

participants had varied periods of exposure to the intervention, a factor that may have affected the estimate of the effect. Fifth, we did not determine compliance to antiepileptic drugs or have reports of adverse effects patients experienced while on treatment. We also did not have a detailed documentation of the nutritional therapy and the cognitive stimulatory activities each child received and did not assess the effect of home environment on outcome. We, however, limited the effects of such bias by choosing only few and fairly robust outcome measures.

Failure to conduct a prospective study means that we cannot comment on the incidence of death or on patients who might have discontinued follow-up care (eg, due to a deterioration in symptoms, severe motor disability or loss of faith in the treatment) leading to an overestimate of the effect. Such an effect, if any, is most likely minimal. From the Ministry of Health epidemiological surveillance reports, only 12 patients with probable nodding syndrome died over the period of observation, mostly from seizure-related events.

Furthermore, our comparative group—participants with other convulsive epilepsies—was a heterogeneous group with different seizure types and possibly neuropathology, on treatment with different anticonvulsants, each with different efficacy, dose and side effects. It would have served us better to recruit a more homogeneous group of patients, for example, only patients with generalised seizures on treatment with a single anticonvulsant. However, in this rural community, the diagnosis of epilepsy is only limited to clinical features obtained on history and clinical observations by clinicians with limited training. Despite this weakness, our results clearly demonstrate that the outcome of nodding syndrome is different from that of the combined heterogeneous group of patients with the other convulsive epilepsies.

### Conclusions

The symptoms and psychomotor functioning of patients with nodding syndrome improve with symptomatic treatments suggesting that nodding syndrome is probably a reversible epileptic encephalopathy. Symptom reversibility may depend on the timing of interventions. Uncontrolled epileptic seizures may be a major contributor to the neurocognitive decline and disability in this syndrome. Further studies are recommended to elucidate these findings.

**Author affiliations**
[1]Department of Paediatrics and Child Health, Mulago Hospital/Makerere University College of Health Sciences, Kampala, Uganda
[2]Nuffield Department of Medicine, Centre for Tropical Medicine and Global Health, University of Oxford, Oxford, UK
[3]Department of Psychiatry, Mulago Hospital/Makerere University College of Health Sciences, Kampala, Uganda
[4]Division of Child and Adolescent Psychiatry, Red Cross War Memorial Children's Hospital, University of Cape Town, South Africa
[5]Butabika National Referral Hospital, Kampala, Uganda

[6]Astrid Lindgren Children's Hospital, Neuropediatric Research Unit, Karolinska Institutet, Sweden
[7]Department of Internal Medicine, Mulago Hospital/Makerere University College of Health Sciences, Kampala, Uganda
[8]Ministry of Health Headquarters, Kampala, Uganda

**Acknowledgements** The authors would like to thank the Local District Health Services in Lamwo, Kitgum, Pader, Amuru, Gulu and Lira who made the study possible; Keron Ssebyala, who managed the data; and Professor Gretchen Birbeck, who critically reviewed the first draft of the this manuscript and offered helpful comments to improve it.

**Contributors** RI, HN, CA, BBM, AK-M, ROO, AKM, ADM and BTO designed the intervention and the study and participated in supervising patient care. RI and HN performed the data analysis. RI wrote the first draft and all participated in data interpretation and provided a critical review of the manuscript.

**Funding** The study was funded by the Government of Uganda. RI is partly supported by the Wellcome Trust through a Director's discretionary research funds.

**Competing interests** None.

**Ethics approval** Makerere University School of Medicine Research and Ethics Committee.

**Provenance and peer review** Not commissioned; externally peer reviewed.

**Data sharing statement** No additional data are available.

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
