## [Reviewer comments · BMJ Open]

ARTICLE DETAILS

TITLE (PROVISIONAL)	Nodding syndrome patients in Uganda improve with symptomatic treatment; a cross sectional study
AUTHORS	Idro, Richard; Namusoke, Hanifa; Abbo, Catherine; Mutamba, Brian; Kakooza, Angelina; Opoka, Robert; Musubire, Abdu; Mwaka, Amos; Opar, Bernard

VERSION 1 - REVIEW

REVIEWER	Erich Schmutzhard Department of Neurology Medical University Innsbruck Austria
REVIEW RETURNED	07-Sep-2014

GENERAL COMMENTS	ad2) abstract: in the abstract a brief explanation, what exactly symptomatic treatment means, needs to be included. I understood that this symptomatic tx was much more than only giving anticonvulsants. beside this minor point on the abstract (see above) I would - in addition - suggest to 1) to give clear information on the duration of the respective disease (i.e. mean duration of NS and mean duration of OCE)2) similarly, it is absolutely essential to know the exact percentages and intensities of all comorbidities (e.g. malnutrition, history of injuries, burns; other chronic diseases (e.g. concomitant tuberculosis, malignancies (e.g. Burkitt)), h.o. severe or, even better, cerebral malaria, acute meningitis or encephalitis, diarrheal episodes etc.). With this in line would be to have the best possible information on vaccination status of these children/adolescents
--

REVIEWER	Andrea S Winkler Klinikum rechts der Isar, Technical University of Munich Germany
REVIEW RETURNED	22-Sep-2014

GENERAL COMMENTS	This study describes the effect of antiepileptic treatment as well as emotional and behavioural interventions together with physiotherapy and nutritional support in children with nodding syndrome compared to children with other convulsive epilepsy disorders. The study represents an important analysis in the research of nodding syndrome. I have some comments that should be addressed before publication:
---

Introduction:

First paragraph in the introduction (page 4 line 5-7): please make better use of the literature.

1. The authors should quote Sejvar et al. 2013 and Foltz et al. 2013 for Uganda, Nyungura et al. 2011, Tumwine et al 2012 and Spencer et al. 2013 for South Sudan and Winkler 2008, 2010 and 2014 for Tanzania. I would not quote Lacey, Kaiser and Edwards as this is not a scientific papers, but reports or answers to one of the other scientific articles.

2. Talking about the number of affected people the latest data from the community-based prevalence study published lately in MMWR needs to be quoted.

Methods:

Page 7, the last two paragraphs: Please specify what was compared with what. You say that data was collected between July and September 2014. Which point in time was this data compared to? Were you able to compare to exactly 12 month prior? How often did the children come to the clinic? On a monthly basis? According to my own working experience in Africa, it would be very difficult to compare it to retrospective data as clinical notes are often fairly incomplete. How was this potential shortcoming accounted for?

Page 8, last paragraph: The authors state that “The outcomes of patients with nodding syndrome were then compared to those of patients with other convulsive epilepsies.” However, throughout the result section nodding children and OCE children are statistically compared within their own group over time. There does not seem to be a comparison between the two groups.

Discussion:

Page 12, second last paragraph: “The improvements we observed were however less than those seen in patients with other convulsive epilepsies suggesting that epileptic seizures in nodding syndrome may be less anticonvulsant sensitive...” I have great difficulties with this statement and the control group per se and am not sure whether the control group is really contributing a great deal to the main outcome of the paper. The control group consists of children with different seizure types, reflecting different neuropathology (children with generalized and (!) focal seizures were included). In addition, these children were on different anticonvulsants and different doses with different anticonvulsant efficacies and obviously side effects. In a nutshell, the control group is extremely heterogenous and it is rather unclear why not a more homogenous group of children with e.g. generalized seizures treated with phenobarbitone was chosen. The advantages and disadvantages of the control group would need in-depth discussion within the discussion section.

Page 12, last paragraph:” This observation seems to concur with a report from Tanzania where symptoms of nodding syndrome completely resolved in four of the original cohort of 62 patients.” Two of these four patients were off treatment, so this does not really reflect response to treatment. The authors should compare their results with those patients treated with phenobarbitone in the same patient cohort published in the same paper.

VERSION 1 – AUTHOR RESPONSE

Reviewer 1

1. In the abstract, a brief explanation on what exactly symptomatic treatment means, needs to be included. I understood that this symptomatic treatment was much more than only giving anticonvulsants.

The introduction section of the abstract has been amended to provide the brief explanation as requested although this was limited by the required length of the abstract.

2. Give clear information on the duration of the respective disease (i.e. mean duration of NS and mean duration of OCE)

This information had already been provided in the results section of the manuscript as the median duration of symptoms (the data is skewed). This is now also included in the abstract.

3. Similarly, it is essential to know the exact percentages and intensities of all co-morbidities (e.g. malnutrition, history of injuries, burns; other chronic diseases (e.g. concomitant tuberculosis, malignancies (e.g. Burkitt)), h.o. severe or, even better, cerebral malaria, acute meningitis or encephalitis, diarrheal episodes etc.). With this in line would be to have the best possible information on vaccination status of these children/adolescents.

We regret to say that we are only able to provide some of this information. With poor record keeping and limited health records available especially with the massive internal displacement that occurred in this area as a result of the Lord's Resistance Army insurgency, the risk of inaccurate data and therefore bias would be great if we attempted to obtain this information from patient recall. This concern has now been addressed in the discussion section as a study limitation.

Reviewer 2

This study describes the effect of antiepileptic treatment as well as emotional and behavioural interventions together with physiotherapy and nutritional support in children with nodding syndrome compared to children with other convulsive epilepsy disorders. The study represents an important analysis in the research of nodding syndrome.

1. First paragraph in the introduction (page 4 line 5-7): please make better use of the literature. The authors should quote Sejvar et al. 2013 and Foltz et al. 2013 for Uganda, Nyungura et al. 2011, Tumwine et al 2012 and Spencer et al. 2013 for South Sudan and Winkler 2008, 2010 and 2014 for Tanzania. I would not quote Lacey, Kaiser and Edwards as this is not a scientific papers, but reports or answers to one of the other scientific articles.

Thank you for the comments. These changes have been made.

2. Talking about the number of affected people the latest data from the community based prevalence study published lately in MMWR needs to be quoted.

This reference has now been added and quoted under design and setting.

3. Page 7, the last two paragraphs: Please specify what was compared with what.

We compared the pre-and post intervention outcome measures in each of the two groups and also

examined the relationship between seizure control and patient characteristics including duration and age at onset of symptoms, baseline seizure frequency, presence of behaviour and emotional difficulties, whether the child had head nodding only or head nodding plus (other seizures) and antiepileptic drug dose.

4. You say that data was collected between July and September 2014. Which point in time was this data compared to? Were you able to compare to exactly 12 month prior? How often did the children come to the clinic? On a monthly basis? According to my own working experience in Africa, it would be very difficult to compare it to retrospective data as clinical notes are often fairly incomplete. How was this potential shortcoming accounted for?

This information was NOT collected exactly at 12 months but 12 – 15 months after initiation of therapy. We collected data between July and September 2013 depending on when the patient reported for the monthly follow up care but in all cases, all participants had completed at least 12 months of the intervention (range 12 - 15 months). We compared the pre-intervention measures to data obtained at these time points. The varied periods on the intervention may have affected the estimate of the effects. These comments have now been included in the methods section and in the limitations.

As for the retrospective data, we used only limited outcome measures so that most of the measures were available. As indicated in the study profile in figure 1, we excluded patients who did not have these critical measurements and also left out nutritional status because a big proportion of patients did not have height readings taken on initiation of intervention.

5. Page 8, last paragraph: The authors state that “The outcomes of patients with nodding syndrome were then compared to those of patients with other convulsive epilepsies.” However, throughout the result section nodding children and OCE children are statistically compared within their own group over time. There does not seem to be a comparison between the two groups.

We have removed this erroneous statement as no direct comparisons of the two independent data sets were made but only comparisons of the improvements in each group over the year.

6. Page 12, second last paragraph: “The improvements we observed were however less than those seen in patients with other convulsive epilepsies suggesting that epileptic seizures in nodding syndrome may be less anticonvulsant sensitive...” I have great difficulties with this statement and the control group per se and am not sure whether the control group is really contributing a great deal to the main outcome of the paper. The control group consists of children with different seizure types, reflecting different neuropathology (children with generalized and (!) focal seizures were included). In addition, these children were on different anticonvulsants and different doses with different anticonvulsant efficacies and obviously side effects. In a nutshell, the control group is extremely heterogenous and it is rather unclear why not a more homogenous group of children with e.g. generalized seizures treated with phenobarbitone was chosen. The advantages and disadvantages of the control group would need in-depth discussion within the discussion section.

We thank the reviewer for this important observation. Indeed we acknowledge the limitations introduced by the choice of our comparative group. The diagnosis and classification of epilepsy in this rural community is quite limited and in many cases, categorisation into specific clinical groups is not possible. We therefore chose all convulsive epilepsies and only left out non convulsive epilepsies. Although this choice brings with it problems of heterogeneity, it is clear from our results that in general, the outcome of nodding syndrome was worse than that of this heterogeneous group of patients. Therefore, our conclusions still stand. We however now have this weakness put down in the limitations section of revised manuscript. The basis for this choice has also been included in the

methods section under participants.

7. Page 12, last paragraph:” This observation seems to concur with a report from Tanzania where symptoms of nodding syndrome completely resolved in four of the original cohort of 62 patients.” Two of these four patients were off treatment, so this does not really reflect response to treatment. The authors should compare their results with those patients treated with phenobarbitone in the same patient cohort published in the same paper.

This change has been made. The comparison is now between 32 patients (two of who became seizure free on phenobarbitone but the majority of who had reduced seizures) and the comparable group in the current study.

VERSION 2 – REVIEW

REVIEWER	Andrea Winkler Department of Neurology Klinikum rechts der Isar Technical University Munich
REVIEW RETURNED	14-Oct-2014

GENERAL COMMENTS	Thanks for your responses and for acting upon my advices. I have no more concerns and recommend the paper for publication.
--